# Two-step regulation of *trachealess* ensures tight coupling of cell fate with morphogenesis in the *Drosophila* trachea

Takefumi Kondo[1,2]*, Shigeo Hayashi[3,4]*

[1]Graduate School of Biostudies, Kyoto University, Kyoto, Japan; [2]The Keihanshin Consortium for Fostering the Next Generation of Global Leaders in Research (K-CONNEX), Kyoto, Japan; [3]Laboratory for Morphogenetic Signaling, RIKEN Center for Biosystems Dynamics Research, Kobe, Japan; [4]Department of Biology, Kobe University Graduate School of Science, Kobe, Japan

**Abstract** During organogenesis, inductive signals cause cell differentiation and morphogenesis. However, how these phenomena are coordinated to form functional organs is poorly understood. Here, we show that cell differentiation of the *Drosophila* trachea is sequentially determined in two steps and that the second step is synchronous with the invagination of the epithelial sheet. The master gene *trachealess* is dispensable for the initiation of invagination, while it is essential for maintaining the invaginated structure, suggesting that tracheal morphogenesis and differentiation are separately induced. *trachealess* expression starts in bipotential tracheal/epidermal placode cells. After invagination, its expression is maintained in the invaginated cells but is extinguished in the remaining sheet cells. A *trachealess* cis-regulatory module that shows both tracheal enhancer activity and silencer activity in the surface epidermal sheet was identified. We propose that the coupling of *trachealess* expression with the invaginated structure ensures that only invaginated cells canalize robustly into the tracheal fate.
DOI: https://doi.org/10.7554/eLife.45145.001

**\*For correspondence:**
take-kondo@lif.kyoto-u.ac.jp (TK);
shigeo.hayashi@riken.jp (SH)

**Competing interests:** The authors declare that no competing interests exist.

## Introduction

A fundamental question in biology is how cells coordinately shape functional organs with complex architecture during embryogenesis. Extensive studies have uncovered how inductive signals, such as morphogens, prime cell differentiation and morphogenesis (*Heisenberg and Bellaïche, 2013*; *Perrimon et al., 2012*), leading to segregated organs with uniquely specified cells. Due to the graded nature of the inductive signals, the initial territories of an organ primordial placode are occupied by cells with various degrees of commitment. Furthermore, cells modulate their own physical properties by changing gene expression to drive morphogenesis, but each cell behavior is dynamic and fluctuating. Therefore, mechanisms to coordinate these phenomena are of critical importance. Without a coordination mechanism, tissues would be mixed with improperly specified cells that would interfere with organ functions. The sequence of signaling, gene expression and morphogenesis is not unidirectional, and the feedback input from morphogenesis to gene expression is proposed to be crucial (*Chan et al., 2017*; *Gilmour et al., 2017*). However, the generality of the proposed feedback mechanisms from morphogenesis to gene expression and cell differentiation in a wide range of developmental systems remains to be determined.

Epithelial invagination is an important morphogenetic process in which three-dimensional tubular organs are formed from a two-dimensional flat sheet (*Andrew and Ewald, 2010*; *Kondo and Hayashi, 2015*; *Sawyer et al., 2010*), and the *Drosophila* trachea is a useful model system for analyzing three-dimensional epithelial morphogenesis (*Hayashi and Kondo, 2018*; *Loganathan et al., 2016*).

**eLife digest** Cells in developing organs have two important decisions to make: where to be and what cell type to become. If cells end up in the wrong places, they can stop an organ from working, so it is vital that one decision depends upon the other. The so-called progenitor cells responsible for forming the trachea, for example, can either become part of a flat sheet or part of a tube. The cells on the sheet need to become epidermal cells, while the cells in the tube need to become tracheal cells. Work on fruit flies found that a gene called '*trachealess*' plays an important role in this process. Without it, developing flies cannot make a trachea at all.

At the start of trachea development, some of the cells form thickened structures called placodes. The progenitor cells in the placodes start to divide, and the structures buckle inwards to form pockets. These pockets then lengthen into tubes. The *trachealess* gene codes for a protein that works as a genetic switch. It turns other genes on or off, helping the progenitor cells inside the pockets to become tracheal cells. But, it is not clear whether *trachealess* drives the formation of the pockets: the progenitor cells first decide what to be; or whether pocket formation tells the cells to use *trachealess*: the progenitor cells first decide where to be.

To find out, Kondo and Hayashi imaged developing fly embryos and saw that the *trachealess* gene does not start pocket formation, but that it is essential to maintain the pockets. Flies without the gene managed to form pockets, but they did not last long. Looking at embryos with defects in other genes involved in pocket formation revealed why. In these flies, some of the progenitor cells using *trachealess* got left behind when the pockets started to form. But rather than forming pockets of their own (as they might if *trachealess* were driving pocket formation), they turned their *trachealess* gene off. Progenitor cells in the fly trachea seem to decide where to be before they decide what cell type to become. This helps to make sure that trachea cells do not form in the wrong places.

A question that still remains is how do the cells know when they are inside a pocket? It is possible that the cells are sensing different mechanical forces or different chemical signals. Further research could help scientists to understand how organs form in living animals, and how they might better recreate that process in the laboratory.

DOI: https://doi.org/10.7554/eLife.45145.002

Tracheal morphogenesis is initiated by placode specification; ten pairs of tracheal placodes form in the dorsal anterior part of the epidermis in each segment by stage 10, followed by invagination, branching and fusion (*Figure 1A*). In this process, the tracheal placodes first appear as a group of cells expressing *trachealess* (*trh*), which is considered to be a master regulator of tracheal morphogenesis (*Chung et al., 2011*; *Isaac and Andrew, 1996*; *Wilk et al., 1996*), and then EGF signaling and mitosis synergistically drive invagination by generating centripetal pressure and inducing epithelial sheet buckling, respectively (*Kondo and Hayashi, 2013*; *Nishimura et al., 2007*; *Ogura et al., 2018*). Finally, FGF signaling triggers tracheal branching (*Figure 1A*) (*Glazer and Shilo, 1991*; *Sutherland et al., 1996*).

*trh* encodes a bHLH-PAS transcription factor that is critical for tracheal morphogenesis. Its expression is primarily induced under the combinatorial control of activation through JAK-STAT signaling and inhibition through Wg and Dpp signaling before invagination (*Brown et al., 2001*; *Wilk et al., 1996*), and STAT-responsive enhancers for *trh* have been identified (*Sotillos et al., 2010*). After invagination, all of the tracheal cells continue expressing *trh*, while no other surrounding epithelial cells, such as epidermal cells, express this factor. However, it is not well understood how *trh* expression is strictly restricted only to invaginated tracheal cells. Although *trh* is proposed to maintain its own expression through an auto-regulatory mechanism (*Wilk et al., 1996*; *Zelzer and Shilo, 2000*), it is still unclear whether all the cells that start expressing Trh expression take part in the invagination and generation of trachea, or if some of these cells fail to invaginate, and if so, how they shut off the auto-regulatory control of *trh*.

In this article, we first show that *trh* plays a critical role in maintaining the invaginated structure but not in initiating invagination. Second, we reveal that the tracheal placode cells initiating *trh* expression later become either tracheal or epidermal cells, and the maintenance of *trh* expression is

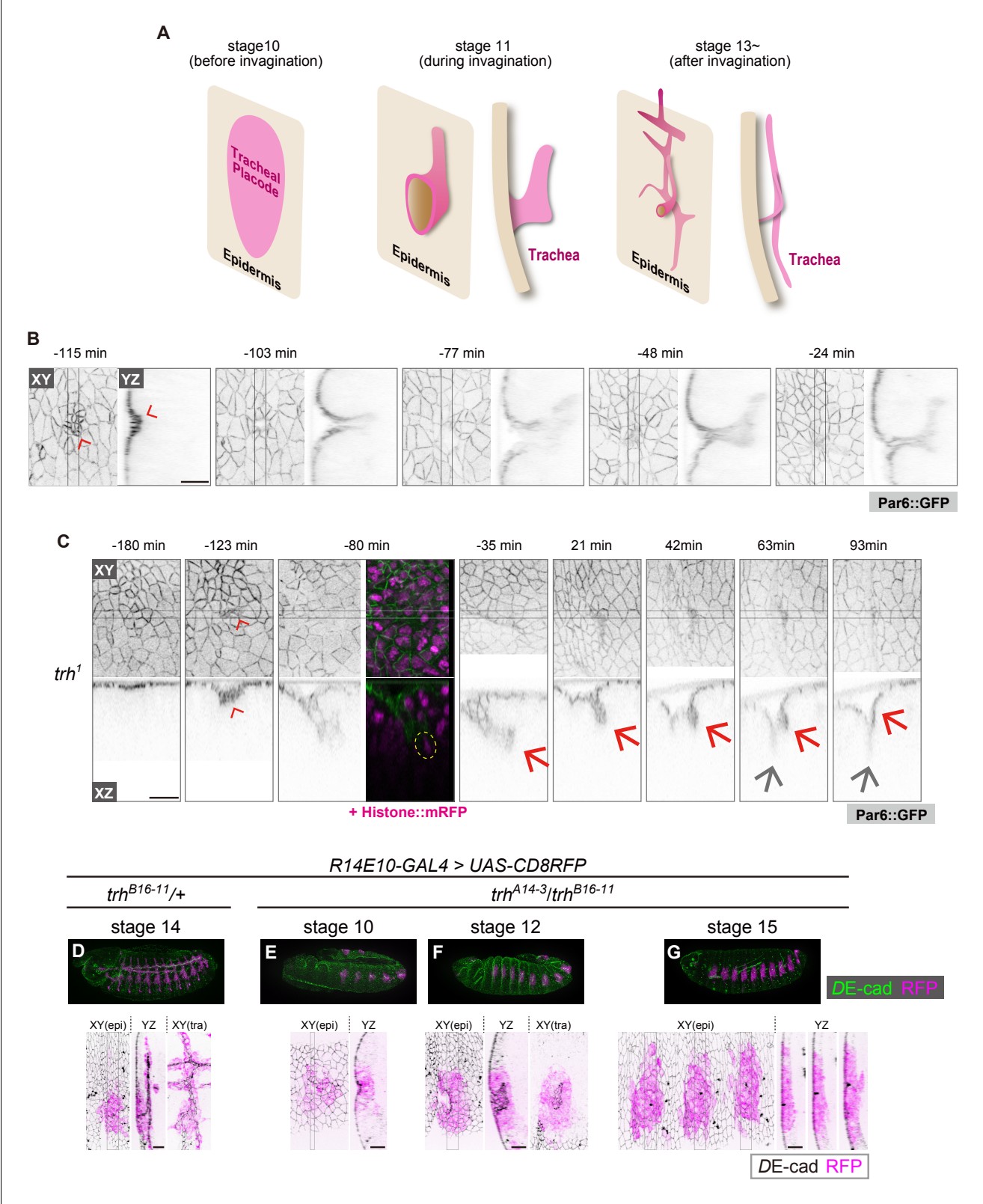

**Figure 1.** *trh* is essential for maintaining the invaginated tracheal structures. (**A**) Schematic of the tracheal morphogenesis process. For clarity, only apical surfaces are shown. (**B, C**) Live imaging of tracheal invagination in a control embryo (**B**) and a *trh* mutant (**C**). Red arrowheads: apical constriction forming a tracheal pit. Yellow circle: a mitotic cell associated with accelerated invagination, distinguished by condensed histone. Red arrows: transient invagination and return to epidermis in a *trh1* mutant. Gray arrows: segmental groove, which is not a tracheal structure. Par-6::GFP indicates the apical

*Figure 1 continued on next page*

*Figure 1 continued*

cell side, and His2Av::mRFP indicates chromosomes. Time point zero is set to the onset of germband retraction. (**D**) Activity of *R14E10-GAL4* in a control embryo monitored using *UAS-mCD8RFP*. (**E–G**) Activity of *R14E10-GAL4* in *trh*[A14-3/B16-11] mutant embryos monitored using *UAS-mCD8RFP*. Green: *DE-cad*, Magenta: mCD8RFP driven by *R14E10-GAL4*. Cells expressing RFP initiated invagination at stage 10 (**E**), and invaginated structures formed within the RFP-positive cell cluster at stage 12 (**F**). However, these invaginated structures were not observed at stage 16, and RFP-positive cells were observed in the surface epidermis (**G**). Scale bars, 10 μm.

DOI: https://doi.org/10.7554/eLife.45145.003

The following figure supplements are available for figure 1:

**Figure supplement 1.** Activation pattern of EGFR signaling before invagination.

DOI: https://doi.org/10.7554/eLife.45145.004

**Figure supplement 2.** Characterization of TALEN-induced *trh* mutants.

DOI: https://doi.org/10.7554/eLife.45145.005

**Figure supplement 3.** The *trh* mutant phenotype was not rescued by *btl* overexpression or the inhibition of apoptosis.

DOI: https://doi.org/10.7554/eLife.45145.006

tightly associated with the change from the epithelial sheet to the invaginated structure. On the basis of our findings, we propose that the transcriptional coordination of *trh* expression, tracheal cell fate specification and invaginated structures during epithelial invagination ensures that only the invaginated cells are canalized robustly into the tracheal fate.

## Results

### *trh* is required to maintain invaginated structures

We previously reported that mitosis can drive tracheal invagination, alone or in combination with EGFR signaling (*Kondo and Hayashi, 2013*). Although all embryonic cells undergo multiple cell divisions, mitosis-induced invagination occurs only in the tracheal placode. Non-tracheal epidermal cells quickly recover their flat epithelial architecture after mitosis, suggesting that the tracheal placode cells possess a special ability to couple mitosis with invagination and tubule formation (*Kondo and Hayashi, 2013*). Since *trh* is considered a master regulator of tracheal morphogenesis (*Isaac and Andrew, 1996*; *Wilk et al., 1996*), we reasoned that *trh* is involved in this mitosis-induced invagination.

Previous studies showed that in *trh* mutants, the tracheal tissue is completely missing in late-stage embryos, and no invagination occurs (*Isaac and Andrew, 1996*; *Wilk et al., 1996*; *Younossi-Hartenstein and Hartenstein, 1993*). However, in the stage-10 tracheal placode, di-phosphorylated ERK, a hallmark of EGFR activation, was detected even in *trh* mutants (*Figure 1—figure supplement 1*) (*Ogura et al., 2018*), suggesting that some early tracheal development processes were taking place. Live imaging of *trh*[1] (an EMS-induced missense allele) mutants at single-cell resolution revealed an unexpected finding: apical constriction forming a tracheal pit appeared in the center of the placode region, followed by mitosis in the pit cells and rapid, deep invagination as seen in the control, although the onset of invagination was delayed (*Figure 1B,C*). Over the next 90 min, the invaginated structure gradually returned to the surface epidermal layer and merged with these cells to form a segmental furrow, leaving no trace of the tracheal structure (*Figure 1C*). Consistent with this live imaging analysis of *trh*[1], in fixed samples with a heteroallelic combination of TALEN-induced *trh* null alleles (*trh*[A14-3]/*trh*[B16-11], *Figure 1—figure supplement 2*) (*Kondo et al., 2014*), would-be tracheal placode cells labeled with the *R14E10 trh* primary enhancer (containing the *trh66* STAT-responsive element that mediates stage-10 *trh* expression in the placode (*Sotillos et al., 2010*), *Figure 1D*) also formed a tracheal pit (*Figure 1E*) and formed invaginated structures during stages 11–13 (*Figure 1F*). These *R14E10*-positive cells returned to the surface epidermal layer at stage 15 (*Figure 1G*). The appearance of invaginated structures at stage 12 and the disappearance of these structures at stages 15–16 were observed, as shown in *Figure 1F and G*, respectively, with 100% penetrance (stage 12: eight embryos, stage 15–16: ten embryos). In addition, the overexpression of *trh* (*trh*-OE) by *R14E10-GAL4* in the *trh* mutants rescued the phenotype (*Figure 1—figure supplement 3A*). One of the Trh target genes is *breathless* (*btl*), which encodes an FGF receptor (*Ohshiro and Saigo, 1997*), and we reported that FGF signaling through Btl is able to

trigger invagination independent of EGF signaling and mitotic rounding (*Kondo and Hayashi, 2013*). However, the *btl*-OE in the *trh* mutants using *R14E10-GAL4* did not rescue tracheal formation from the transiently invaginated tracheal placodes (*Figure 1—figure supplement 3B*), indicating that other *trh* target genes are required to support FGF signaling-triggered tracheal morphogenesis. In addition, inhibiting apoptosis by *p35* did not prevent the transiently invaginated cells from returning to the epidermis, indicating that apoptotic cell removal is not the major cause of this anomaly (*Figure 1—figure supplement 3C*). These results demonstrated that *trh* is essential for maintaining the invaginated structure, whereas it is dispensable for initiating invagination. Thus, tracheal formation proceeds by two successive and genetically separable steps: (1) invagination triggered by the mechanical forces generated through the combined activities of mitosis, EGFR, and FGFR signaling; and (2) maintenance of the invaginated structure controlled by *trh*. Since mitosis-triggered invagination was maintained in *rho bnl* mutants (*Kondo and Hayashi, 2013*), EGFR and FGFR signaling are dispensable for the maintenance of the invaginated structure.

## Only invaginated tubule cells maintained Trh expression

For Trh to function as a determinant of the invaginated structures, its expression must be tightly sustained only in the invaginated cells but not in the surface epidermal cells. To reveal the relationship between *trh* expression and epithelial geometry, we attempted to analyze the impact of reducing the number of invaginated tracheal cells on Trh expression. In *rho bnl* double mutants that lose both EGF and FGF signaling in tracheal cells, tracheal invagination is impaired, and the trachea is composed of a smaller number of cells than that of the control. If a similar number of cells initiates *trh* expression in control and *rho bnl* mutants, some *trh*-positive placode cells are expected to remain in the surface epidermis, and these cells may face a conflict between their fate and tissue geometry.

JAK-STAT signaling induces *trh* expression through a STAT-responsive *trh* enhancer in the tracheal placodes at stage 10 of embryogenesis before invagination (*Figure 2A*) (*Brown et al., 2001*; *Sotillos et al., 2010*). After invagination, Trh is detected only in all invaginated cells, including the most proximal spiracular branch (*Figure 2C,F*). The number of initial Trh+ cells in the tracheal placode before invagination (stage 10) was 58.2 ± 5.1 (mean ±S.D.) in controls and 67.2 ± 8.5 in the *rho bnl* mutants, respectively, indicating that tracheal fate specification was not compromised in the *rho bnl* mutants (*Figure 2A,B,G*). The increase in the number of initial Trh+ cells in the *rho bnl* placodes reflects the expansion of the tracheal placode due to the earlier role of *rho* in restricting the size of tracheal placode (*Raz and Shilo, 1993*). The *R14E10* fragment contains the *trh66* STAT-responsive element that mediates stage9-10 *trh* expression (*Sotillos et al., 2010*), and the number of initial *R14E10* + cells (57.9 ± 2.7 cells, monitored by using the *R14E10-lacZ* transgene at stage 10) is almost the same as the number of initial Trh+ cells. In contrast, after cycle-16 mitosis and invagination, the resultant tracheae of the *rho bnl* mutants were composed of a smaller number of Trh+ cells (*trh*-on cells, 31.2 ± 6.2 cells) than those of the controls (87.6 ± 6.3 cells) (*Figure 2C,D,G*). These findings demonstrate that in the *rho bnl* mutants, the number of initial Trh+ cells at stage 10 was reduced at stages 13–14. This reduction is due to either the disappearance of Trh+ cells from the epithelium or the loss of Trh expression in cells that failed to invaginate.

To discriminate these possibilities, we traced the fate of cells initiating *trh* expression by labeling them with *nls-lacZ* driven by *R14E10*. The *R14E10*-GAL4-induced *nls-lacZ* product (β-galactosidase, β-gal) persisted in the initial *trh+* cells after termination of *R14E10-GAL4* transcription, allowing us to distinguish *trh*-off (Trh⁻, LacZ⁺) and *trh*-on (Trh⁺) cells derived from the initial *trh*-on cell population after invagination (from stage 13 onward). Even in control embryos, there were *trh*-off cells (31.9 ± 6.7 cells) in the epidermis (*Figure 2C,G*), while all of the *trh*-on cells were found in the invaginated tubule region (87.6 ± 6.3 cells). The sum of the *trh*-on and *trh*-off cells (119.5 ± 10.5 cells) agreed well with the prediction from the number of initial Trh+ cells (58.2 ± 5.1 cells) and the number of initial R14E10 + cells (57.9 ± 2.7 cells, *Figure 3—figure supplement 2B,C*) after one round of cycle-16 mitosis during invagination. The results showed that 27% of the initial *trh+* cells lost their Trh expression, all of which remained in the epidermis. Many of the *trh*-off cells remained during the rest of embryogenesis and formed trichomes on their apical surface (data not shown), suggesting that they adopted the epidermal fate.

We next asked if the loss of *trh* expression in *trh*-off cells was due to their failure to become part of the tube by tracing the fate of *trh*-expressing cells in the *rho bnl* mutants. After invagination, the resultant tracheae were composed of a smaller number of *trh*-on cells (31.2 ± 6.2 cells), as

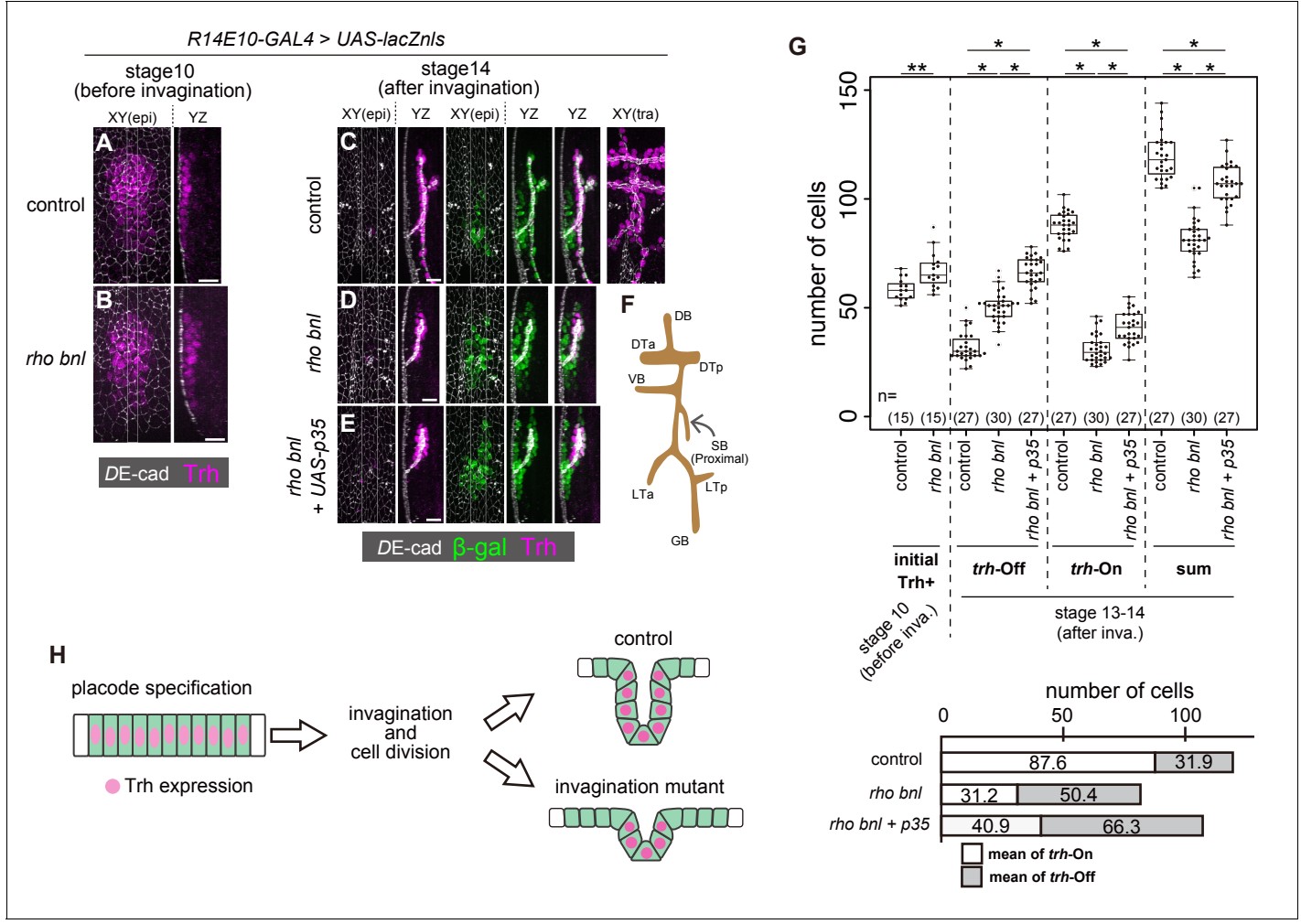

**Figure 2.** Trh expression is maintained only in invaginated tracheal cells. (A, B) Trh expression in a tracheal placode of a control ($rho^{del1}$ $bnl^{P1}$ /+) (a) and a $rho^{del1}$ $bnl^{P1}$ mutant (B) embryo at stage 10 before invagination. (C–E) Trh and β-gal expression in a control embryo (C), a $rho^{del1}$ $bnl^{P1}$ mutant (D), and a $rho^{del1}$ $bnl^{P1}$ mutant with p35 overexpression (E) at stage 14 after invagination. β-gal and p35 expression were driven by R14E10-GAL4. (F) Schematic of the tracheal branching pattern after invagination. (G) Upper: Boxplot of cell numbers. Initial Trh+: the number of Trh-expressing cells before invagination, trh-Off: the number of cells expressing β-gal driven by R14E10-GAL4 in the epidermis (Trh-negative) after invagination (stage 13–14), trh-On: the number of invaginated tracheal cells expressing Trh after invagination (stage 13–14), sum: the sum of trh-Off and trh-On. **: Exact Wilcoxon-Mann-Whitney Test, p=0.001331, *: Steel-Dwass test, p<0.001 (for trh-Off, control vs rho bnl: $p=5.3 \times 10^{-9}$, control vs rho bnl +p35: $p=8.3 \times 10^{-10}$, rho bnl vs rho bnl +p35: $p=3.9 \times 10^{-8}$; for trh-On, control vs rho bnl: $p=5.28 \times 10^{-10}$, control vs rho bnl +p35: $p=8.4 \times 10^{-10}$, rho bnl vs rho bnl +p35: $p=2.8 \times 10^{-10}$; for sum, control vs rho bnl: $p=3.1 \times 10^{-10}$, control vs rho bnl +p35: $p=2.0 \times 10^{-4}$, rho bnl vs rho bnl +p35: $p=3.2 \times 10^{-9}$). Lower: mean numbers of trh-On cells and trh-Off cells at stages 13–14 after invagination. (H) Schematic of the dynamics of Trh expression during invagination. Scale bars, 10 μm.

DOI: https://doi.org/10.7554/eLife.45145.007

The following source data and figure supplement are available for figure 2:

**Source data 1.** Source data for *Figure 2G* and *Figure 2—figure supplement 1B*.
DOI: https://doi.org/10.7554/eLife.45145.009
**Figure supplement 1.** Only a few invaginated cells maintain Trh expression in *rho CycA bnl* mutants.
DOI: https://doi.org/10.7554/eLife.45145.008

mentioned above, and surrounded by an increased number of epidermal *trh-off* cells (50.4 ± 7.2 cells) than those of the controls, with a total of 81.6 ± 9.8 cells (*Figure 2D,G*). Blocking apoptosis by *p35* in the *rho bnl* mutants increased the number of both *trh-on* and *trh-off* cells (*Figure 2E,G*). In the *rho bnl* mutants with or without p35, 62% of the surviving initial *trh+* cells remained in the epidermis and lost their *trh* expression. These findings indicated that the reduction in *trh-on* cells in the

*rho bnl* mutants was not simply due to their disappearance from the epithelium. Losses of EGF, FGF, and mitosis in the *rho CyclinA (CycA) bnl* triple mutant caused a more severe invagination defect (*Kondo and Hayashi, 2013*). The initial *trh* expression at stage 10 was nearly normal even in the triple mutant (53.9 ± 7.1 cells (*Figure 2—figure supplement 1A,B*). If the cells maintaining Trh expression and forming tubes are predetermined before invagination, the triple mutants are supposed to possess half the number of *trh*-on cells observed in the *rho bnl* double mutants because the *CycA* mutation eliminates cycle-16 mitosis. However, although all the invaginated cells were Trh-positive, the number of Trh-on tracheal cells in these triple mutants was much smaller than expected (6.0 ± 2.5 cells, *Figure 2—figure supplement 1A,B*), which strongly argues against the model in which the tube-forming *trh*-on cells are predetermined before invagination. These observations support the possibility that the *trh* expression in the stage 10 tracheal placodes is maintained only in the successfully invaginated tubule cells, independent of the depth of invagination. The placode cells that failed to invaginate and remained in the epidermis lost their *trh* expression. These results imply that a mechanism exists to maintain *trh* expression only in the invaginated tubule cells and extinguish it in the superficial epidermal cells (*Figure 2H*).

## *R15F01* is the *trh* enhancer that is sensitive to changes in tissue geometry

A transcriptional reporter of *trh* [1-eve-1, a *lacZ* enhancer trap of *trh* (*Perrimon et al., 1991*; *Wilk et al., 1996*) elicited reporter β-gal expression that was limited to the invaginated tracheal cells (*Figure 3B*), suggesting that the epidermal expression of *trh* is repressed at the level of transcription. We then tested the properties of several previously identified *trh* enhancers (*Sotillos et al., 2010*). Among the eight *trh* upstream regions with phylogenetically conserved STAT-binding sites, *trh47* and *trh66* drive reporter expression from the early stage of tracheal development, and *trh67* is proposed to be a *trh*-dependent auto-regulatory element (*Sotillos et al., 2010*). We found that the two primary enhancers, *trh47* (*Figure 3A*, *Figure 3—figure supplement 1A*) and *trh66* (covered by *R14E10*, *Figure 3A*, *Figures 1D* and *2C*), were active in both tubule cells and the surrounding epidermal cells, suggesting that they did not reproduce the epidermal extinction of *trh*. In contrast, *trh67* did not drive reporter expression in all of the Trh-positive invaginated cells, suggesting that additional *cis*-elements control the tube-specific maintenance of *trh* expression (*Figure 3—figure supplement 1B*).

We then searched for additional *trh* enhancers from a systematic enhancer mapping resource (the FlyLight project: https://www.janelia.org/project-team/flylight) (*Jenett et al., 2012*; *Jory et al., 2012*; *Manning et al., 2012*) and identified another enhancer immediately upstream of the proximal *trh* promoter (*R15F01*, *Figure 3A*). *R15F01* showed strict tube-specific activity after invagination, and its activity was maintained in the invaginated tubule cells throughout embryogenesis (*Figure 3B', G*) and in postembryonic stages (not shown), in contrast to *trh47* and *trh66* (*R14E10*), which show transient activity only from embryonic stage 10 to 11 (*Sotillos et al., 2010*). We note that a few epidermal cells sporadically showed leaked *R15F01* activity, especially when we monitored the activity using *R15F01-GAL4* reporter transgenes (*Figure 3B'*). In addition, when we used a direct lacZ reporter (*R15F01-lacZ*), *R15F01*'s activity became detectable in part of the tracheal placode before invagination (*Figure 3F*, *Figure 3—figure supplement 2G*), slightly later than that of the other early enhancers (*trh47* or *R14E10*). In addition, we noticed that *R15F01* repressed the function of an adjacent *mini-yellow* gene (*mini-y* included in *attP40 or attP2*, a transgene landing site on chromosome 2L or chromosome 3L, respectively [*Groth et al., 2004*]) in the adult epidermis (*Figure 3E* and *Figure 3—figure supplement 2A*). The function of the endogenous *yellow* gene on chromosome X was not affected (data not shown), suggesting that *R15F01* represses *mini-y* expression in cis. These results suggested that *R15F01* is a cis-regulatory module (CRM) that simultaneously functions as an enhancer in tracheal tube cells and a silencer in epidermal sheet cells.

Using the mutant combinations that prevented invagination to various degrees (combinations of *rho, bnl* and *CycA*), no or only a few epidermal cells showed *R15F01* activity, while the small tracheal tubes were *R15F01*-positive irrespective of the depth of invagination (*Figure 3C,D,H–K*). In the *rho bnl* mutants, *R15F01* activation was detected at stage 10 before invagination. The number of initial *R15F01* + cells in the *rho bnl* mutants before cycle-16 mitosis and invagination (39.5 ± 4.7 cells, *Figure 3—figure supplement 2D,G*) is smaller than that of control embryos (47.3 ± 6.1, *Figure 3—figure supplement 2G*), indicating that EGF signaling is involved in the initial activation but is not

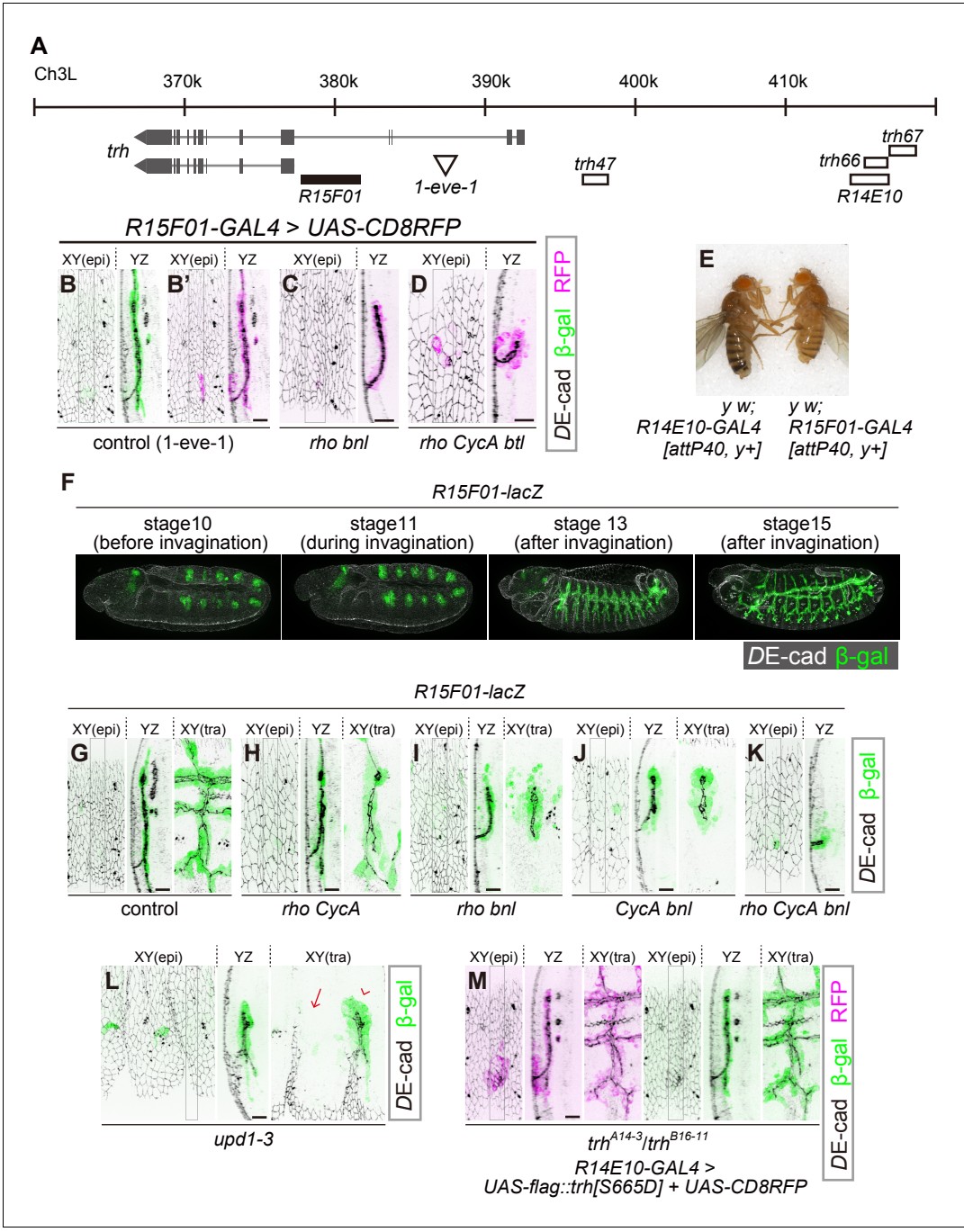

**Figure 3.** *R15F10* reproduces the tubule-restricted Trh pattern. (**A**) Genomic positions of *R15F01*, other enhancers, and the insertion site of *1-eve-1* at the *trh* locus (**B–D**) Enhancer activity of *R15F01* monitored using *R15F01-GAL4* with *UAS-mCD8RFP* in a control (**B'**), *rho^{del1} bnl^{P1}* mutant (**C**), and *rho^{del1} CycA^{C8LR1} btl^{deltaOh10}* mutant (**D**) embryo. β-gal indicates the expression from a lacZ enhancer trap line for *trh*, *1-eve-1* (**B**). (**E**) Phenotype in adult cuticle pigmentation. The *R15F01-GAL4* fly had a lighter body color than the *R14E10-GAL4* fly, indicating that *R15F01* represses an adjacent *mini-yellow* gene. (**F–M**) Enhancer activity of *R15F01* monitored using the direct *lacZ* reporter in control embryos and in several invagination mutants: *rho^{del1} CycA^{C8LR1}* (**H**), *rho^{del1} bnl^{P1}* (**I**), *CycA^{C8LR1} bnl^{P1}* (**J**), *rho^{del1} CycA^{C8LR1} bnl^{P1}* (**K**), *Df(1)BSC352* (deficient in all *upd1, 2* and *3*), arrowhead: a segment with invaginated trachea, arrow: a segment without trachea (**L**), and *trh^{A14-3/B16-11} R14E10-GAL4 > UAS-flag::trh[S665D]* (**M**). Scale bars, 10 μm.

DOI: https://doi.org/10.7554/eLife.45145.010

The following source data and figure supplements are available for figure 3:

*Figure 3 continued on next page*

*Figure 3 continued*

**Figure supplement 1.** Activity of *trh* enhancers in tracheal cells and surrounding epidermal cells.
DOI: https://doi.org/10.7554/eLife.45145.011
**Figure supplement 2.** Activity of *R14E10* and *R15F01* in *rho bnl* and *trh* mutants.
DOI: https://doi.org/10.7554/eLife.45145.012
**Figure supplement 2—source data 1.** Source data for *Figure 3—figure supplement 2C,G,H*.
DOI: https://doi.org/10.7554/eLife.45145.013
**Figure supplement 3.** *trh*-overexpression by using R15F01-GAL4 rescued the *trh* mutant phenotype in maintaining invaginated structures.
DOI: https://doi.org/10.7554/eLife.45145.014

essential. In addition, the number is larger than the number of invaginated cells with Trh expression after cycle-16 mitosis (Trh-on cells) in the *rho bnl* mutants (two times $37.7 \pm 6.3$ cells VS *trh*-on cells $31.2 \pm 6.2$ cells (*Figure 2G*); note that the number of cells is doubled after cycle-16 mitosis), suggesting that *R15F01* activity is controlled by a multistep mechanism. Even in *unpaired 1* (*upd1*), *upd2*, and *upd3* triple mutants (*Df(1)BSC352*), which are deficient for all Upd cytokines and are unable to activate JAK-STAT signaling, invaginated trachea appeared in some segments, and only these invaginated cells became *R15F01*-active (*Figure 3L*). These results indicate that the tubule-specific maintenance of *R15F01* activity after invagination is independent of JAK-STAT, EGFR, and FGFR signaling. These results also support the idea that the tube-forming *trh*-on cells are not predetermined before invagination and that only the successfully invaginated cells secondarily sustain the *R15F01* activity for Trh expression independent of the initial induction of its activation.

To analyze the contribution of *trh* to tube-specific *R15F01* maintenance, we monitored *R15F01* activity in *trh* mutants. *R15F01* activation was detected in the *trh* mutants before invagination, but the number of β-gal-positive cells ($29.4 \pm 6.7$) was smaller than that of the control ($47.3 \pm 6.1$) (*Figure 3—figure supplement 2E,G*), indicating that *trh* is involved in the initial activation of *R15F01* but is not essential. After mitosis cycle 16, transient invagination and disappearance of invaginated architecture in the *trh* mutants, $19.1 \pm 6.8$ cells still maintained *R15F01* reporter expression at stage 15 (*Figure 3—figure supplement 2F,H*). Because this number is smaller than the number of initially activated cells, *R15F01* activity could be maintained in a *trh*-independent manner. As mentioned above, in addition, *trh*-OE in the *trh* mutants by *R14E10-GAL4* (early transient activation of *trh* in placodes, and no secondary regulation of *trh* expression) could rescue tracheal morphogenesis, and more cells took on a tubular architecture than in the *trh* mutants (*Figure 1—figure supplement 3A*). We found that all these invaginated cells showed *R15F01* activity (*Figure 3M*, β-gal), while some *trh*-OE cells did not take part in invagination, and these surface-remaining cells did not show *R15F01* reporter expression even when we overexpressed a phosphomimetic active form of Trh (*Figure 3M*, RFP + epidermal cells). These results strongly suggest that Trh is not sufficient to maintain longer *R15F01* activity and that secondary regulatory mechanisms that are potentially associated with invagination are required. *trh*-OE by *R15F01-GAL4* in the *trh* mutants could also partially rescue tracheal morphogenesis (*Figure 3—figure supplement 3*). Because of the later onset of *R15F01* and/or the smaller number of *R15F01*-active cells than in *R14E10* (*Figure 3—figure supplement 2*), the invagination defect of the *trh* mutants was not rescued at stage 12. On the other hand, in later stage 15, *R15F01*-positive cells were able to maintain invaginated structures, although they showed an incomplete branching pattern, possibly due to a limited number of *R15F01*-positive cells.

## The *R15F01* CRM possesses both tracheal enhancers and epidermal silencers

To identify functional elements within the 3963 bp fragment of *R15F01*, we divided *R15F01* into eight fragments (D1-D8), constructed four deletions (del1-del4) and assayed their regulatory activity before invagination (stage-10 tracheal placode), their activity in the trachea and epidermis after invagination (from stage 13 onward), and their *cis*-inhibition effect on *mini-yellow* in adult flies (*Figure 4A*). The results showed that the enhancer activity for tracheal expression was mapped to two sub-fragments: D7, which was sufficient to drive expression in the tracheal placode and

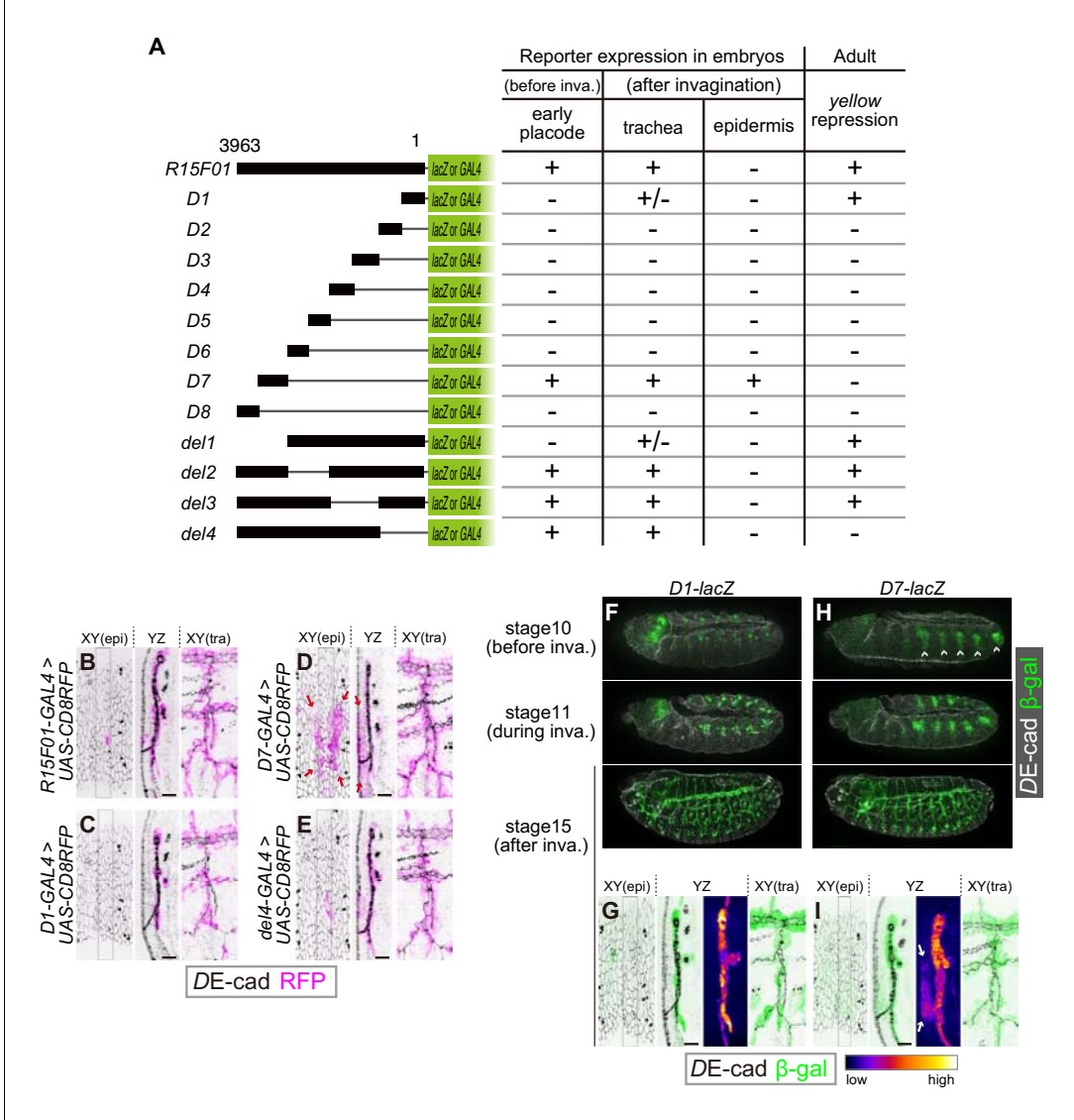

**Figure 4.** The *R15F10* CRM contains multiple tracheal enhancers and epidermal silencers. (A) Summary of the domain mapping of *R15F01*. ± indicates sporadic expression. (B–E) Activities of *R15F01* (full length), D1, D7, and del4 monitored using GAL4 and *UAS-mCD8RFP*. D7 activated RFP expression in both tracheal and surrounding epidermal cells (arrows in E). D1 induced RFP expression in a sporadic manner. (F–I) Activities of D1 and D7 monitored using the direct *lacZ* reporter at stage 10 (before invagination), stage 11 (during invagination), and stage 15 (after invagination). D7 showed enhancer activity in the tracheal placodes before invagination, but D1 activated the reporter in tracheal cells after invagination. Arrows in I indicate epidermal expression of β-gal. Scale bars, 10 μm.

DOI: https://doi.org/10.7554/eLife.45145.015

The following figure supplement is available for figure 4:

**Figure supplement 1.** Enhancer and silencer activities of truncated *R15F01* elements.

DOI: https://doi.org/10.7554/eLife.45145.016

invaginated tracheal tubules (*Figure 4A,D,H,I*), and D1, which drove tracheal expression after invagination in a somewhat sporadic manner (*Figure 4A,C,F,G*). No other sub-fragment showed tracheal enhancer activity (*Figure 4—figure supplement 1B–G*). In addition, fragment D7 also drove expression in the epidermal region near the tracheal pit (*Figure 4D,I*), indicating that D7 drives both tracheal and epidermal expression and is not sufficient to reproduce the tube-restricted pattern after invagination. Second, the *mini-y* silencer activity was mapped to sub-fragment D1, and its removal from *R15F01* (i.e., the del4 construct) abrogated this *mini-yellow* silencing (*Figure 4A*, *Figure 4—figure supplement 1A*).

The epidermal enhancer activity of D7 in embryos was not detected in full-length *R15F01,* suggesting that an epidermal silencer that represses D7's epidermal activity must reside in another part of *R15F01*. D1 is likely to contain this activity, since it possesses the silencer activity for *mini-yellow*, and the combination of D1 with D7 in constructs del2 and del3 reproduced the expression pattern of *R15F01* (*Figure 4A*, *Figure 4—figure supplement 1I,J*). In addition, the del4 construct, which did not contain the D1 or D2 fragment, also showed epidermal suppression of D7 activity (*Figure 4A,E*, *Figure 4—figure supplement 1K*). These data indicated that silencer elements reside in at least two regions, one in the D1-D2 fragment with *mini-y* silencer activity and the other in the region included in del4 (in the D3 to D6 and/or D8 fragment).

To examine whether these silencers could act on a heterologous epidermal enhancer, we constructed chimeric reporters consisting of the *shavenbaby* (*svb*) epidermal enhancer and sub-fragments of *R15F01* (*Figure 5A*). The *svb* E6B element enhances reporter gene expression in the dorsal epidermal cells that form trichomes (*Figure 5B,D*) (*Frankel et al., 2011*). We found that the fragments containing D1 and D2 (D1-D2), from D3 to D6 (D3-D6), and from D1 to D6 (in the del1 construct) were able to silence the *svb* E6B activity (*Figure 5C,E,F*). Since the D1-D2 and D3-D6 inserts have no overlap, these data confirmed that *R15F01* contains multiple and redundant silencer elements that act dominantly over the *svb* E6B epidermal enhancers.

To further characterize the difference in the silencing activity of *R15F01* between epidermal and internal epithelial tissues, we constructed another chimeric reporter in which the UAS element was fused to *R15F01*. When we crossed the 3×UAS-GFP reporter with arm-GAL4, which is an epithelial ubiquitous GAL4 driver (*Sanson et al., 1996*), GFP expression was detected ubiquitously, with an intense signal in epidermal and hindgut cells (*Figure 5G*). Then, we crossed the 3×UAS-*R15F01*-fused GFP reporter with arm-GAL4, and epidermal GFP expression was nearly undetectable, while hindgut GFP expression was still detectable (*Figure 5H*). The tracheal GFP signal was enhanced, possibly due to the tracheal enhancers in *R15F01*. Embryos harboring only the 3×UAS-*R15F01* reporter without arm-GAL4 showed the tracheal GFP signal but not hindgut GFP expression (*Figure 5I*), indicating that the hindgut activity was driven by arm-GAL4. These results are also consistent with the notion that *R15F01* silences enhancer activities in the surface epidermis but not in internal tubular organs.

## Discussion

Here, we showed that during *Drosophila* tracheal morphogenesis, (1) the master regulator *trh* is essential for maintaining the invaginated structure of the trachea but is dispensable for driving invagination of the placode, and (2) *trh* expression is maintained in invaginated cells, while its expression in surface epidermal cells is actively repressed. We propose that under these two mechanisms, the only successfully invaginated cells establish tight coupling between different hierarchies, tracheal cell fate and tubular architecture. *trh*-positive placode cells that do not take part in tubules lose their *trh* expression and adopt the epidermal fate with a flat sheet architecture.

### Driving forces of morphogenesis and the stable structures of epithelial tissue

We found that in *trh* mutants, a subset of the would-be tracheal placode cells undergo invagination but fail to maintain the invaginated structure. This observation contradicts previous reports claiming that *trh* is required for invagination (*Isaac and Andrew, 1996*; *Wilk et al., 1996*). We consider that our results based on live imaging of early tracheal invagination processes identified crucial tracheal cell behavior that was missed in previous works that mainly focused on late embryonic phenotypes. The phenotype of *trh* mutants indicates that the conversion of the epithelial sheet of the tracheal placode into a tube through invagination and the stabilization of the invaginated structures are genetically separable steps. In addition, inductive signals, such as JAK/STAT signaling, are considered to prime both tracheal differentiation (i.e., *trh* expression) and invagination separably. This finding is also consistent with the idea that morphogenetic movement and cell differentiation can be uncoupled (*Ip et al., 1994*). We suggest that epithelial tissue can assume two alternative stable structures, sheet or tube, and according to cell fate, each epithelial tissue assumes one of these structures. In the tracheal system, invagination forces include the contraction of myosin cables regulated by EGF signaling and a cell migratory force stimulated by FGF signaling (*Kondo and Hayashi,*

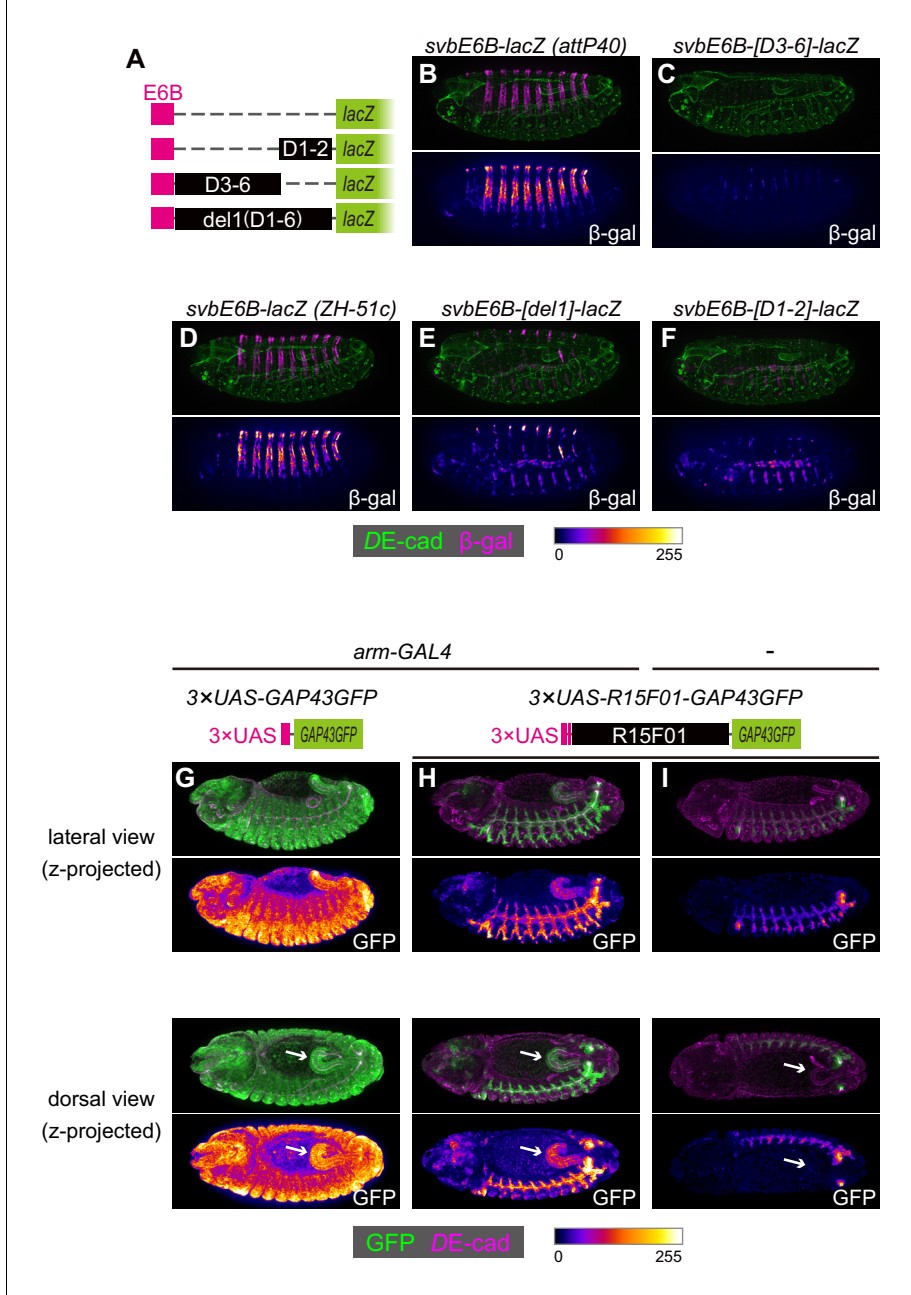

**Figure 5.** The *R15F01* epidermal silencers counteract a heterologous epidermal enhancer. (**A**) Schematic of chimeric reporters with *svb-E6B* and *R15F01* fragments (**B, C**) Reporter β-gal expression (magenta or fire) from *E6B-lacZ* and *E6B-[D3-6]-lacZ* integrated at the *attP40* site. The epidermal β-gal expression in *E6B-[D3-6]-lacZ* was significantly weaker than that in *E6B-lacZ*. (**D–F**) Reporter β-gal expression (magenta or fire) from *E6B-lacZ*, *E6B-[D1-2]-lacZ*, and *E6B-[del1]-lacZ* integrated at the *ZH-51c* site. The epidermal β-gal expression in *E6B-[D1-2]-lacZ* and *E6B-[del1]-lacZ* was significantly weaker than that in *E6B-lacZ*, while both showed tracheal reporter expression. (**G–I**) Reporter GFP expression (green or fire) in (**G**) *arm-GAL4 >3×UAS-GAP43GFP*, (**H**) *arm-GAL4 >3 × UAS-R15F01-GAP43GFP*, and (**I**) *3×UAS-GAP43GFP* only. Both *3×UAS-GAP43GFP* and *3×UAS-R15F01-GAP43GFP* transgenes were integrated at the *attP2* site. The epidermal GFP expression (the surface of embryos) in *arm-GAL4 >3×UAS-R15F01-GAP43GFP* was significantly weaker than that of *3×UAS-GAP43GFP*, while hindgut GFP expression was detectable. Embryos possessing only *3×UAS-GAP43GFP* showed tracheal GFP expression but not hindgut expression. Arrows indicate the hindgut.

DOI: https://doi.org/10.7554/eLife.45145.017

*2013*; *Nishimura et al., 2007*; *Ogura et al., 2018*). If both signaling pathways are absent, transient tissue instability caused by clustered mitosis allows invagination (*Kondo and Hayashi, 2013*). This mitotic cue is sufficient for the conversion of the Trh + placodes from the sheet state to the more stable tube state. Once invaginated by any driving forces, the tracheal cells robustly maintain the invaginated structure under the control of Trh.

In *trh* mutants, although the placode cells are able to initiate invagination, the degree of invagination is much smaller than that of the control. This is consistent with our recent report that *trh* controls the propagation of EGFR activation but not the initial activation of EGFR in the placodes (*Ogura et al., 2018*), indicating that *trh* contributes to tracheal invagination through EGFR signaling propagation in part. However, even when both EGFR and FGFR signaling are lost, Trh + cells are able to maintain the invaginated structure. In addition, although *btl,* which encodes an FGFR, is one of the important downstream genes of *trh* and FGFR signaling can trigger invagination when EGFR signaling and mitosis 16 are eliminated in the placodes, the *btl*-OE in the *trh* mutants was not sufficient to rescue invagination and tubule maintenance. These results indicate that the maintenance of the invaginated structure is largely dependent on *trh* in an EGFR and FGFR signaling-independent manner.

While canonical tissue-folding processes are driven by apically concentrated myosin through active apical constriction (*Martin and Goldstein, 2014*), tracheal invagination has unique properties, including passive apical constriction under centripetal pressure from neighboring cells, the acceleration of the invagination through mitotic cell rounding (*Kondo and Hayashi, 2013*), and a lack of apically concentrated myosin localization in the invaginating cells (*Kondo and Hayashi, 2013*). This is consistent with a recent report that myosin regulatory light chain depletion using the deGradFP system does not cause significant abnormalities in tracheal morphogenesis (*Ochoa-Espinosa et al., 2017*). Candidates for Trh-downstream effectors that maintain invaginated structures are Crossvein-less-c (cv-c), a Rho family GTPase-activating protein (RhoGAP), and Crumbs (Crb), both of which are expressed in the tracheal cells under the control of *trh* (*Brodu and Casanova, 2006*; *Letizia et al., 2011*; *Röper, 2012*). However, both *cv-c* mutants and *crb* mutants are able to maintain their tubular tracheal geometry (*Cela and Llimargas, 2006*; *Letizia et al., 2011*), suggesting that *trh* controls the epithelial tissue geometry through the activation of multiple genes. It has recently been reported that in the salivary gland, the overexpression of a constitutive active form of Arp2/3 activator causes reversal of invaginated structures into epidermis (*Chung et al., 2017*), suggesting that the difference in F-actin organization is also important to stabilize epithelial structure (sheet or tube). In addition, *trh* is also known to re-organize microtubule structures (*Brodu et al., 2010*). During dorsal fold formation in the gastrulating fly embryo, the invaginating cells do not show apical myosin enrichment, whereas the apical microtubule network plays an important role in cell shortening through a polarity-dependent basal shift of AJs (*Takeda et al., 2018*; *Wang et al., 2012*). Therefore, re-organization of both F-actin and microtubule architectures, but not apically concentrated myosin, might synergistically support the maintenance of invaginated structures. Further investigation of the *trh*-downstream transcriptome profile and tubule stabilization mechanisms will be important for understanding the diversity of cellular mechanisms of epithelial morphogenesis.

## Induction, priming, and completion of tracheal cell differentiation coupled with epithelial morphogenesis

We found that *trh* expression is strictly maintained only in invaginated cells, whereas it is extinguished in cells remaining in the adjacent surface epidermis, irrespective of the depth of invagination. Because some of the initial Trh-expressing cells do not invaginate and lose their Trh expression, the initial priming of Trh expression is not sufficient for cells to take part in invagination and does not result in autocatalytic maintenance of its expression. Therefore, there should be a mechanism that determines cells that maintain Trh expression after the initial primed state. One possibility is that these cells are predetermined independent of initial priming of *trh* expression before invagination and then form the tubular architecture precisely. The other possibility is that maintenance of Trh expression is tightly associated with invagination. Our data using various invagination mutants (especially the comparison between *rho bnl* mutants and *rho CycA bnl* mutants, *Figure 2* and *Figure 2—figure supplement 1*, as mentioned in the results section above) strongly suggest that the cells that retain *trh* expression and invaginated structures are not predetermined, and tissue-geometry-dependent mechanisms are involved in tight coupling of invaginated structures and Trh expression. Since

Trh is essential for stabilizing invaginated structures, this coupling may ensure that only invaginated cells canalize robustly into the tracheal fate and further supports the formation of a tubular tracheal system after invagination.

Our findings strongly suggest that the *R15F01* CRM is crucial for invagination-restricted Trh expression and is composed of multiple enhancers and redundant epidermal silencers. We also note that because *R15F01* is activated in part of the placode cells before invagination, there should be a mechanism for this transient activation, regardless of tissue architecture. After invagination, *R15F01* activity is maintained only in invaginated cells independent of the degree of invagination, suggesting that *R15F01* senses both initial placode activation cues and morphogenetic invagination cues for *trh* expression. In addition, *R15F01* is also reported to be a conserved Polycomb response element (PRE) (*Hauenschild et al., 2008*; *Schuettengruber et al., 2014*). ChIP signals for several Polycomb factors, such as Polycomb, Pleiohomeotic, Polyhomeotic distal, and Dorsal switch protein 1, are highly concentrated along the *R15F01* region, especially in the D1 region, which shows strong silencer activity for *mini-y*, and this pattern is conserved across *Drosophila* species (*Schuettengruber et al., 2014*) (ChIP atlas; http://chip-atlas.org and GSE60428). Since these analyzes were performed using whole embryos, it is still unclear in which cells these Polycomb factors associate with the *R15F01* locus. It was also recently reported that some developmental enhancers can also function as PREs (*Erceg et al., 2017*). These findings suggest that *R15F01* functions as a developmental enhancer in tracheal cells, while it operates as a PRE in other cells, including epidermal cells. If so, counteracting this PRE activity at the *trh* locus only in invaginated cells may be the critical step for coupling Trh expression and tubular architecture.

## Control mechanisms of *R15F01* CRM activity and Trh maintenance

The terminal differentiation of tracheal cells is likely to be a consequence of the relocation of tracheal primordial cells from the surface epidermis to the inside of the trachea and the suppression of the epidermal silencer activity of *trh*. An essential remaining question is how the tracheal cells couple Trh expression with invaginated structures during morphogenesis. This study showed that known signaling pathways involved in early tracheal morphogenesis, such as JAK-STAT, EGFR and FGFR signaling, are dispensable for maintaining *R15F01* activity only in invaginated cells. One possible mechanism is sensing the change in epithelial geometry from sheet to tube through mechano-transduction pathways. Cells are known to sense rigidity in their environment, mechanical stress, and their own morphology and cytoskeletal architecture and to control gene expression and chromatin organization in response to these factors (*Chan et al., 2017*; *Kirby and Lammerding, 2018*; *Kumar et al., 2017*; *Panciera et al., 2017*). Another possibility is that the cell can detect geometrical conversion through a change in the local concentration of secreted molecules (*Gilmour et al., 2017*). Buckling and bending of the intestinal epithelium affect the local concentration of Sonic hedgehog (Shh) to help define the positions of stem cells in chicks (*Shyer et al., 2015*). In addition, lumen formation and the luminal accumulation of FGF promote differentiation at the zebrafish lateral line (*Durdu et al., 2014*). At present, we do not have evidence that any signaling pathways affected by secreted ligands or known mechano-transduction pathways, such as Hippo, Src-Arm, or $Ca^{2+}$, are involved in invagination-responsive Trh expression. In addition, it is still unclear whether cells directory sense the invaginated structure to terminally differentiate into tracheal cells or whether the relocation from the surface epidermis to the inside of the embryos just allows cells to acquire a non-epidermal fate. It remains a future challenge to discriminate among these possibilities.

Because all tissues and organs must match their architecture with their cellular phenotype to function properly, similar canalization mechanisms that couple gene expression, cell fate, and tissue geometry may play fundamental roles in shaping functional organs. Although morphogenetic feedback is proposed to be important for organogenesis, the cellular mechanisms for sensing tissue geometry have only begun to be elucidated. Further study into the cellular and genetic mechanisms by which tracheal cells monitor the process of morphogenesis and adjust their cell fate would help us understand the robustness of animal morphogenesis.

# Materials and methods

## Key resources table

| Reagent type (species) or resource | Designation | Source or reference | Identifiers | Additional information |
|---|---|---|---|---|
| Genetic reagent (*Drosophila melanogaster*) | trh[1] | Kyoto stock center | DGRC:106845; FLYB: FBal0017036 | |
| Genetic reagent (*Drosophila melanogaster*) | trh[A14-3] | *Kondo et al., 2014* | FLYB: FBal0344676 | |
| Genetic reagent (*Drosophila melanogaster*) | trh[B16-11] | *Kondo et al., 2014* | FLYB: FBal0344695 | |
| Genetic reagent (*Drosophila melanogaster*) | 1-eve-1 | Bloomington Drosophila Stock Center | BDSC: 8744; FLYB: FBti0002897 | FlyBase symbol: P{ET-L}trh-1-eve-1 |
| Genetic reagent (*Drosophila melanogaster*) | R14E10-GAL4 (attP2) | Bloomington Drosophila Stock Center | BDSC: 48641 | |
| Genetic reagent (*Drosophila melanogaster*) | R15F01-GAL4 (attP2) | Bloomington Drosophila Stock Center | BDSC: 45071; FLYB: FBti0133347 | FlyBase symbol: P{GMR15F01-GAL4}attP2 |
| Genetic reagent (*Drosophila melanogaster*) | UAS-mCD8.ChRFP | Bloomington Drosophila Stock Center | BDSC: 27392; FLYB: FBti0115769 | FlyBase symbol: P{UAS-mCD8.ChRFP}3 |
| Genetic reagent (*Drosophila melanogaster*) | UAS-nls-lacZ | Kyoto stock center | DGRC:108782; FLYB: FBti0002781 | FlyBase symbol: P{UAS-GFP::lacZ.nls}30.1 |
| Genetic reagent (*Drosophila melanogaster*) | UAS-flag::trh | PMID: 11740943 | FLYB: FBal0150204 | |
| Genetic reagent (*Drosophila melanogaster*) | UAS-flag::trh [S665D] | PMID: 11740943 | FLYB: FBal0150205 | |
| Genetic reagent (*Drosophila melanogaster*) | UAS-btl::GFP | Bloomington Drosophila Stock Center | BDSC: 41802; FLYB: FBti0148917 | FlyBase symbol: P{UAS-btl::GFP-S65T}3 |
| Genetic reagent (*Drosophila melanogaster*) | UAS-p35 | Kyoto stock center | DGRC: 108018; FLYB:FBti0012594 | FlyBase symbol: P{UAS-p35.H}BH1 |
| Genetic reagent (*Drosophila melanogaster*) | rho[del1] | PMID: 2110920 | FLYB: FBal0017860 | |
| Genetic reagent (*Drosophila melanogaster*) | bnl[P1] | Bloomington Drosophila Stock Center | BDSC: 6384; FLYB: FBal0057745 | |
| Genetic reagent (*Drosophila melanogaster*) | CycA[C8LR1] | Bloomington Drosophila Stock Center | BDSC: 6627; FLYB: FBal0065308 | |
| Genetic reagent (*Drosophila melanogaster*) | btl[deltaOh10] | *Ohshiro and Saigo, 1997* | FLYB: FBal0083056 | |
| Genetic reagent (*Drosophila melanogaster*) | Df(1)BSC352 | Bloomington Drosophila Stock Center | BDSC: 24376; FLYB: FBab0045128 | |
| Genetic reagent (*Drosophila melanogaster*) | Par6::GFP | PMID: 18854163 | FLYB: FBal0243990 | |

*Continued on next page*

*Continued*

| Reagent type (species) or resource | Designation | Source or reference | Identifiers | Additional information |
|---|---|---|---|---|
| Genetic reagent (*Drosophila melanogaster*) | His2Av::mRFP | Bloomington Drosophila Stock Center | BDSC: 23651; FLYB: FBti0077845 | FlyBase symbol: P{His2Av-mRFP1}II.2 |
| Genetic reagent (*Drosophila melanogaster*) | His2Av::mRFP | Bloomington Drosophila Stock Center | BDSC: 23650; FLYB: FBti0077846 | FlyBase symbol: P{His2Av-mRFP1}III.1 |
| Genetic reagent (*Drosophila melanogaster*) | arm-GAL4[11] | Kyoto stock center | DGRC:106387; FLYB: FBti0002793 | FlyBase symbol: P{GAL4-arm.S}11 |
| Genetic reagent (*Drosophila melanogaster*) | trh66-lacZ | *Sotillos et al., 2010* | FLYB: FBal0265118 | |
| Genetic reagent (*Drosophila melanogaster*) | R15F01-lacZ (attP40) | this paper | N/A | lacZ reporter with R15F01 fragment |
| Genetic reagent (*Drosophila melanogaster*) | R14E10-lacZ (attP40) | this paper | N/A | lacZ reporter with R14E10 fragment |
| Genetic reagent (*Drosophila melanogaster*) | R15F01-D1-lacZ (attP40) | this paper | N/A | lacZ reporter with R15F01-D1 fragment |
| Genetic reagent (*Drosophila melanogaster*) | R15F01-D2-lacZ (attP40) | this paper | N/A | lacZ reporter with R15F01-D2 fragment |
| Genetic reagent (*Drosophila melanogaster*) | R15F01-D3-lacZ (attP40) | this paper | N/A | lacZ reporter with R15F01-D3 fragment |
| Genetic reagent (*Drosophila melanogaster*) | R15F01-D4-lacZ (attP40) | this paper | N/A | lacZ reporter with R15F01-D4 fragment |
| Genetic reagent (*Drosophila melanogaster*) | R15F01-D5-lacZ (attP40) | this paper | N/A | lacZ reporter with R15F01-D5 fragment |
| Genetic reagent (*Drosophila melanogaster*) | R15F01-D6-lacZ (attP40) | this paper | N/A | lacZ reporter with R15F01-D6 fragment |
| Genetic reagent (*Drosophila melanogaster*) | R15F01-D7-lacZ (attP40) | this paper | N/A | lacZ reporter with R15F01-D7 fragment |
| Genetic reagent (*Drosophila melanogaster*) | R15F01-D8-lacZ (attP40) | this paper | N/A | lacZ reporter with R15F01-D8 fragment |
| Genetic reagent (*Drosophila melanogaster*) | R15F01-del1-lacZ(ZH-51C) | this paper | N/A | lacZ reporter with R15F01-del1 fragment |
| Genetic reagent (*Drosophila melanogaster*) | R15F01-del1-lacZ[FS](attP40) | this paper | N/A | lacZ reporter with R15F01-del1 fragment, but lacZ CDS contains a frameshift mutation |
| Genetic reagent (*Drosophila melanogaster*) | R15F01-del2-lacZ (attP40) | this paper | N/A | lacZ reporter with R15F01-del2 fragment |
| Genetic reagent (*Drosophila melanogaster*) | R15F01-del3-lacZ (attP40) | this paper | N/A | lacZ reporter with R15F01-del3 fragment |

*Continued on next page*

*Continued*

| Reagent type (species) or resource | Designation | Source or reference | Identifiers | Additional information |
|---|---|---|---|---|
| Genetic reagent (*Drosophila melanogaster*) | R15F01-del4-lacZ (attP40) | this paper | N/A | lacZ reporter with R15F01-del4 fragment |
| Genetic reagent (*Drosophila melanogaster*) | trh47-GAL4(attP2) | this paper | N/A | GAL4 transgene with trh47 regulatory region |
| Genetic reagent (*Drosophila melanogaster*) | trh67-GAL4(attP2) | this paper | N/A | GAL4 transgene with trh67 regulatory region |
| Genetic reagent (*Drosophila melanogaster*) | R15F01-GAL4(attP40) | this paper | N/A | GAL4 transgene with R15F01 fragment |
| Genetic reagent (*Drosophila melanogaster*) | R14E10-GAL4(attP40) | this paper | N/A | GAL4 transgene with R14E10 fragment |
| Genetic reagent (*Drosophila melanogaster*) | R15F01-D1-GAL4 (attP2) | this paper | N/A | GAL4 transgene with R15F01-D1 fragment |
| Genetic reagent (*Drosophila melanogaster*) | R15F01-D7-GAL4 (attP2) | this paper | N/A | GAL4 transgene with R15F01-D7 fragment |
| Genetic reagent (*Drosophila melanogaster*) | R15F01-del4-GAL4 (attP2) | this paper | N/A | GAL4 transgene with R15F01-del4 fragment |
| Genetic reagent (*Drosophila melanogaster*) | svbE6B-lacZ (attP40) | this paper | N/A | lacZ reporter with svbE6B regulatory region, integrated into attP40 |
| Genetic reagent (*Drosophila melanogaster*) | svbE6B-D3-6-lacZ (attP40) | this paper | N/A | lacZ reporter with svbE6B-R15F01-D3-6 fusion fragment |
| Genetic reagent (*Drosophila melanogaster*) | svbE6B-lacZ (ZH-51C) | this paper | N/A | lacZ reporter with svbE6B regulatory region, integrated into ZH-51C |
| Genetic reagent (*Drosophila melanogaster*) | svbE6B-D1-2-lacZ (ZH-51C) | this paper | N/A | lacZ reporter with svbE6B-R15F01-D1-2 fusion fragment |
| Genetic reagent (*Drosophila melanogaster*) | svbE6B-del1-lacZ (ZH-51C) | this paper | N/A | lacZ reporter with svbE6B-R15F01-del1 fusion fragment |
| Genetic reagent (*Drosophila melanogaster*) | 3×UAS-GAP43GFP (attP2) | this paper | N/A | GFP reporter with 3 × UAS sequences |
| Genetic reagent (*Drosophila melanogaster*) | 3×UAS-R15F01-GAP43GFP (attP2) | this paper | N/A | GFP reporter with 3 × UAS-R15F01 fusion |
| Antibody | anti-β-galactosidase | MP Biomedicals | 55976, RRID:AB_2334934 | rabbit polyclonal, 1:5000 for IHC |
| Antibody | anti-β-galactosidase | Abcam | ab9361, RRID:AB_307210 | chick polyclonal, 1:1000 for IHC |
| Antibody | anti-DE-cad | Developmental Studies Hybridoma Bank | DSHB: DCAD2, RRID:AB_528120 | rat monoclonal, 1:20 for IHC |
| Antibody | anti-RFP | MBL | M155-3, RRID:AB_1278880 | mouse monoclonal, 1:300 for IHC |
| Antibody | anti-DsRed | BD Biosciences | 632397 | rabbit polyclonal, 1:5000 for IHC |

*Continued on next page*

*Continued*

| Reagent type (species) or resource | Designation | Source or reference | Identifiers | Additional information |
|---|---|---|---|---|
| Antibody | anti-GFP | Molecular Probes | A-11122, RRID:AB_221569 | rabbit polyclonal, 1:300 for IHC |
| Antibody | anti-Trh | other | | rabbit polyclonal, 1:100 for IHC, Dr Jordi Casanova (IRB Barcelona) |
| Antibody | Anti-MAP Kinase, Activated | Signa-Aldrich | M8159, RRID:AB_477245 | mouse monoclonal, 1:1000 for IHC |
| Antibody | anti-rabbit IgG Alexa Fluor Plus 488 | Molecular Probes | A-32731, RRID:AB_2633280 | 1:300 for IHC |
| Antibody | anti-rabbit IgG Alexa Fluor Plus 555 | Molecular Probes | A-32732, RRID:AB_2633281 | 1:300 for IHC |
| Antibody | anti-mouse IgG Alexa Fluor Plus 488 | Molecular Probes | A-32723, RRID:AB_2633275 | 1:300 for IHC |
| Antibody | anti-mouse IgG Alexa Fluor Plus 555 | Molecular Probes | A-32727, RRID:AB_2633276 | 1:300 for IHC |
| Antibody | anti-rabbit IgG Alexa Fluor 488 | Molecular Probes | A-11034, RRID:AB_2576217 | 1:300 for IHC |
| Antibody | anti-rabbit IgG Alexa Fluor 555 | Molecular Probes | A-21429, RRID:AB_141761 | 1:300 for IHC |
| Antibody | anti-mouse IgG Alexa Fluor 488 | Molecular Probes | A-11029, RRID:AB_138404 | 1:300 for IHC |
| Antibody | anti-mouse IgG Alexa Fluor 555 | Molecular Probes | A-21424, RRID:AB_141780 | 1:300 for IHC |
| Antibody | anti-Rat IgG DyLignt 649 | Jackson ImmunoResearch | 112-495-167 | 1:300 for IHC |
| Antibody | anti-Rat IgG DyLignt 650 | Abcam | ab102263, RRID:AB_10711247 | 1:50 for IHC |
| Antibody | anti-chick IgY Alexa Fluor 488 | Abcam | ab150173 | 1:300 for IHC |
| Antibody | anti-chick IgY Alexa Fluor 555 | Abcam | ab150174 | 1:300 for IHC |
| Antibody | anti-mouse IgG-biotin | Jackson ImmunoResearch | 715-065-151, RRID:AB_2340785 | 1:500 for TSA amplification |
| Commercial assay or kit | VECTASTAIN Universal Elite ABC Kit | Vector Laboratories | PK-6100 | Use Reagent A and Reagent B for TSA amplification |
| Commercial assay or kit | TSA Cyanine 3 System | PerkinElmer | NEL704A001KT | 1:50 for TSA amplification |

## Fly strains

The fly strains used in this study are listed in the Key Resources Table.

## Plasmid construction and transgenesis

Plasmids were constructed using PrimeSTAR max or PrimeSTAR HS (Takara Bio) and an in-fusion PCR cloning kit (Clontech) unless otherwise noted. For pBPGUw-R15F01 and pBPGUw-R14E10, each corresponding region was amplified from genomic DNA, subcloned into pENTR-TOPO, and recombined into pBPGUw [pBPGUw was a gift from Gerald Rubin (Addgene plasmid #17575) (*Pfeiffer et al., 2008*) using LR recombinase. For pBPGUw-lacZ, the lacZ coding sequence (CDS) was amplified from pCaSpeR-hs-lacZ and subcloned into HindIII-digested pBPGUw. For pBPGUw-R15F01-lacZ, the R15F01 fragment was recombined into pBPGUw-lacZ from pENTR-TOPO-R15F01 using LR recombinase. For pBPGUw-(D1 ~D8 or del1 ~del4)-lacZ, each truncated fragment of R15F01 was amplified from pBPGUw-R15F01 and subcloned into AatII/NaeI-digested pBPGUw-R15F01-lacZ. For pBPGUw-(D1, D7, or del4)-GAL4, each truncated fragment of R15F01

was amplified and subcloned into AatII/NaeI-digested pBPGUw-R15F01. For pBPGUw-trh47 or trh67, each fragment (*Sotillos et al., 2010*) was amplified from genomic DNA and subcloned into AatII/NaeI-digested pBPGUw-R14E10. For pBPGUw-svbE6B-(D1-2, D3-6 or D1-6)-lacZ, each chimeric fragment was amplified from genomic DNA and plasmids containing a R15F01 fragment and subcloned into AatII/NaeI-digested pBPGUw-R15F01-lacZ. For pBPGUw-3×UAS-GAP43GFP and pBPGUw-3×UAS-R15F01-GAP43GFP, the GFP CDS with the GAP43 palmitoylation sequence was amplified from pUbi-GAP-CAAX (*Kondo and Hayashi, 2013*) by PCR and subcloned into HindIII-digested pBPGUw (pBPGUw-GAP43GFP). A 3×UAS fragment were generated by annealing two oligo DNAs and subcloned into AatII/NeaI-digested pBPGUw-GAP43GFP. A R15F01 fragment fused to 3×UAS was amplified by PCR and subcloned into AatII/NeaI-digested pBPGUw-GAP43GFP. The primer sequences used in this study are listed in *Supplementary file 1*.

Transgenic strains were generated by φC31-mediated transgene integration into the attP target sites of *attP2*, *attP40*, or *ZH-51C* (*Bischof et al., 2007*; *Groth et al., 2004*) using plasmid DNAs constructed as described above. Plasmid DNA injections were performed in our laboratory or by Best-Gene. Information about DNA constructs using the *attP* landing site is included in the table of fly strains. Most of the *lacZ* reporters were integrated into the *attP40* site, except for *del1-lacZ*, *svbE6B-D1-2-lacZ*, and *svbE6B-D1-6-lacZ*. Although one strain of *del1-lacZ* integrated at the *attP40* site was obtained, it had a frame-shift mutation in the *lacZ* CDS. Therefore, we used this *attP40* line with the frameshift to observe the adult body color (*Figure 4—figure supplement 1A*) and the *ZH-51C* line to analyze the embryonic β-gal expression (*Figure 4—figure supplement 1H*). *svbE6B-D1-2-lacZ* and *svbE6B-D1-6-lacZ* were also integrated into the *ZH-51C* site because we could not obtain transformants in which these transgenes were integrated into the *attP40* site.

## Live imaging

Embryos were prepared for live imaging as previously reported (*Kondo and Hayashi, 2013*). Imaging was performed using an Olympus FV-1000 with a 60x oil immersion objective (PLAPON 60XO, numerical aperture 1.42, Olympus) at 25°C (*Figure 1C*) or a Zeiss LSM800 with a 63x oil immersion objective (Objective Plan-Apochromat 63x/1.4 Oil DIC, Zeiss) (*Figure 1B*) at 25°C with a setting below saturated signal intensity. Images were processed using FIJI software (https://fiji.sc/), and all projection views were generated using the custom FIJI plugin CoordinateShift (written by Housei Wada, https://signaling.riken.jp/en/en-tools/imagej/). 'XY' showed the Z-projection view. 'YZ' and 'XZ' showed the X-projection and Y-projection views of a boxed area in the 'XY' panels. The range of intensity was adjusted using FIJI software, avoiding saturation of the signal.

## Immunohistochemistry

Embryos were dechorionated in 50% bleach for 2 min and fixed in 1:1 4% PFA containing 1 mM $CaCl_2$ and heptane for 20 min at room temperature. The vitelline membrane was removed by shaking in 1:1 methanol and heptane. Embryos were washed in PBSTwx (PBS with 0.2% Tween 20% and 0.2% Triton X-100) 3 times for 15 min each and blocked in PBSTwx with 1% BSA for 60 min at room temperature. Samples were stained with the primary antibody at 4 °C overnight and washed in PBSTwx 3 times for 15 min each. Secondary antibody or phalloidin staining was performed at room temperature for 3 hr. The antibodies used in this study are listed in the Key Resources Table. For dpERK antibody staining, the fluorescent signal was amplified using the Tyramide Signal Amplification system with anti-mouse IgG-biotin, Reagents A and B in the ABC kit, and Cy3 Tyramide. After staining, the embryos were washed in PBSTwx 3 times for 15 min each and mounted in Vectashield Mounting Medium with DAPI (Vector Laboratories) or SlowFade Diamond Antifade Mountant with DAPI (Molecular Probes).

Images of fixed embryos were taken using a Zeiss Apotome.2 equipped with ORCA-Flash V2 (Hamamatsu Photonics) and a 20x dry objective (Objective Plan-Apochromat 20x/0.8, Zeiss) or a 63x water immersion objective (Objective C-Apochromat 63x/1.20 W, Zeiss) with a setting below saturated signal intensity unless otherwise noted. For *Figure 1—figure supplement 1*, images of fixed embryos were taken using an Olympus FV-1000 with a 20x objective lens (UPLSAPO 20X numerical aperture 0.75, Olympus). Images were processed using FIJI software, and all projection views were generated using a custom FIJI plugin CoordinateShift (https://signaling.riken.jp/en/en-tools/imagej/). The Z-projection, X-projection, and Y-projection regions were manually determined for each

image. 'XY(epi)' showed the Z-projection view at the surface-epidermis level, and 'XY(tra)' showed the Z-projection view at the level inside the trachea. 'YZ' and 'XZ' showed the X-projection and Y-projection views of a boxed area in the 'XY' panels. All images were acquired with a 63x water immersion objective (Objective C-Apochromat 63x/1.20 W, Zeiss) were smoothed with a 1-Sigma (radius) Gaussian Blur filter. The dynamic range of intensity was adjusted while avoiding saturation of the signal. For the *DE*-cad signal, the intensity was adjusted to show the epithelial tissue geometry more clearly; therefore, some dense signals were oversaturated. For the *svbE6B* chimeric reporter analyzes in *Figure 5B–F*, the β-gal signal was collected and adjusted using the same parameters among strains using the same transgene landing site (*attP40* or *ZH-51c*). For the $3 \times UAS$ chimeric reporter analyzes in *Figure 5G–I*, the GFP signal was collected and adjusted using the same parameters. All images were converted into 8-bit images and assembled using Adobe Illustrator for figures. In all images, the anterior side is to the left, and the dorsal side is up. Since tracheal morphogenesis is left-right symmetric, right-side images were flipped to adjust the directions of the anterior-posterior and dorsal-ventral axes.

## Cell counting

The numbers of Trh-positive cells and β-gal-positive cells were counted manually using Z-stack images taken by a Zeiss ApoTome.2 and FIJI software with the Cell Counter plug-in. The 4th, 5th, and 6th tracheal metameres of each embryo were used for this quantification. Boxplots and beeswarms were drawn, and statistical analyzes were performed using R software (https://www.r-project.org/). Exact Wilcoxon-Mann-Whitney Tests were performed using the Wilcox_test function from the coin package. Steel-Dwass tests were performed using pSDCFlig from the NSM3 package with the Asymptotic method.

## Adult fly imaging

Images of 1-day-old adult flies were taken using a Leica S8APO stereomicroscope equipped with an Olympus AIR01 digital camera. Images were adjusted using Adobe Photoshop and assembled using Adobe Illustrator for figures.

## Acknowledgements

We thank the Kyoto Stock Center, the Bloomington Drosophila Stock Center, the Developmental Studies Hybridoma Bank, Tetsuya Kojima, James Castelli-Gair Hombría and Jordi Casanova for fly stocks and antibodies; Housei Wada, Yoshimi Takeda, Yu Kurata, and Masayo Miki for assistance with the data analysis and experiments; Tadashi Uemura, Tadao Usui, Yu-Chiun Wang, and Yosuke Ogura for critical comments on the manuscript; and members of the Hayashi, Nishimura, Wang, Kuranaga, and Uemura laboratories for helpful discussion. This work was supported by a Grant-in-Aid for Scientific Research (15H05597 to TK) from the Japan Society for the Promotion of Science (JSPS), Precursory Research for Innovative MEdical care (PRIME) of the Japan Agency for Medical Research and Development (AMED) (JP18gm5810020 to TK), and the Keihanshin Consortium for Fostering the Next Generation of Global Leaders in Research (K-CONNEX) established by the program of Building of Consortia for the Development of Human Resources in Science and Technology, MEXT (to TK).

## Additional information

### Funding

| Funder | Grant reference number | Author |
| --- | --- | --- |
| Japan Agency for Medical Research and Development | JP18gm5810020 | Takefumi Kondo |
| Japan Society for the Promotion of Science | 15H05597 | Takefumi Kondo |

| Ministry of Education, Culture, Sports, Science, and Technology | Building of Consortia for the Development of Human Resources in Science and Technology | Takefumi Kondo |

The funders had no role in study design, data collection and interpretation, or the decision to submit the work for publication.

## Author contributions
Takefumi Kondo, Conceptualization, Data curation, Formal analysis, Funding acquisition, Validation, Investigation, Visualization, Writing—original draft, Project administration, Writing—review and editing; Shigeo Hayashi, Conceptualization, Supervision, Writing—review and editing

## Author ORCIDs
Takefumi Kondo (iD) https://orcid.org/0000-0001-8127-1141
Shigeo Hayashi (iD) https://orcid.org/0000-0001-7785-1290

## Decision letter and Author response
Decision letter https://doi.org/10.7554/eLife.45145.021
Author response https://doi.org/10.7554/eLife.45145.022

# Additional files

## Supplementary files
• Supplementary file 1. List of PCR primers.
DOI: https://doi.org/10.7554/eLife.45145.018
• Transparent reporting form
DOI: https://doi.org/10.7554/eLife.45145.019

## Data availability
Cell counting data are available as Source data files.

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
