## [Decision Letter]

Thank you for submitting your article "Two-step regulation of *trachealess* ensures tight coupling of cell fate with tubular architecture in *Drosophila* trachea" for consideration by *eLife*. Your article has been reviewed by three peer reviewers, one of whom served as a guest Reviewing Editor, and the evaluation has been overseen Utpal Banerjee as the Senior Editor. The reviewers have opted to remain anonymous.

The reviewers have discussed the reviews with one another and the Reviewing Editor has drafted this decision to help you prepare a revised submission.

Summary:

The study described in the submitted manuscript from Kondo and Hayashi explores the genetic and morphological programs underlying the initial developmental stages of the tracheal system in *Drosophila* embryos. Major findings described include:

• The bHLH-PAS transcription factor Trachealess (Trh), long considered a "master regulator" of tracheal system development, is shown to govern tracheal system differentiation only following invagination and segregation of (future) tracheal cells from the surface ectodermal layer, but to be dispensable for the definition of the tracheal domain and the invagination process itself.

• Restriction of Trh activity to differentiating cells is achieved by refining the relatively broad initial expression pattern of *trh*, so that expression is maintained in invaginated cells but extinguished in cells that remain at the surface. The authors identify a 5' regulatory element, *R15F01*, which appears to harbor the sequences critical both for maintenance and silencing of *trh* expression in relevant cells.

• Data is provided in an attempt to support a scenario where Trh governs tracheal system differentiation via regulation of a RhoGTPase-actomyosin cytoskeletal network.

Essential revisions:

This is a valuable and interesting study whose primary message is the coupling of morphogenesis (in this case- invagination of surface epidermis) with the expression pattern of a key transcription factor that governs further tissue differentiation. All three reviewers viewed both the subject matter and the quality of much of the experimental work with high regard. All three, however, raised a number of non-trivial concerns regarding the significance of different results obtained, as well as aspects of the text, which seemed at times excessive in its interpretation of the data. The major critiques and suggested ways of addressing them include the following:

I) The reported results can be divided conceptually into three groups-

1) The role of Trh: the claim that Trh function is required only subsequent to invagination is a key observation at the heart of this study, which also contrasts with current "dogma" in the field. It is imperative therefore that a strong case be made.

• Can the authors demonstrate that invagination followed by retraction is a prevalent phenotype in *trh* mutant embryos (i.e.- how often is it observed)?

• Can this be shown for full nulls (rather than/in addition to the hypomorphic *trh^1^* allele)?

• Finally- the authors should explicitly address and discuss the discrepancies between their observations and those of previous studies in the Results or Discussion sections.

2) Upstream of *trh*: There was general agreement that the strong part of the study was the focus and analysis of how *trh* expression is regulated in invaginated vs. surface cells, and the identification of the *R15F01* regulatory element as the key mediator of this program. Two additional experiments that could support and provide added significance to this notion were requested:

• Assessment of whether driving *trh* expression using *R15F01* alone is sufficient for rescue of the *trh* mutant phenotype.

• Express GFP throughout the embryonic epidermis (e.g. via da-GAL4) via a UAS-based construct which also harbors *R15F01*, and examine whether GFP expression is silenced in the surface epidermal cells and maintained in invaginated cells of nascent salivary glands (an analogous tubular organ system). If this were to be the case, it would considerably strengthen the notion that the enhancing and silencing functions of *R15F01* may indeed be sensitive to "tissue architecture", as claimed in the text.

3) Downstream of *trh*: Two of the reviewers found multiple problems and weaknesses in the sections on downregulation of Rho activity (possibly via Cv-c) and subsequent effects on actomyosin functions as a mechanism via which Trh governs tracheal morphogenesis. These related to the quality and interpretation of the data, and to the lack of coherent cellular and molecular scenarios that could account for formation of a tubular network. Given the extent of these criticisms, it appears premature to include this aspect of the study- certainly in its present state- in the manuscript, and the consensus among the reviewers is to remove it altogether. This would include the second and third sections of the Results. It is recommended, however, that demonstrations showing that invaginated tracheal cells have different expression or activation of markers than the surface epidermis be retained in some fashion, in order to support the notion of distinct fates. The authors could also speculate on possible downstream mechanisms on the basis of the Trh target list they compiled (Figure 3—figure supplement 2).

This shortening of the text should not detract from the significance of the remaining portions of the study. Characterization of the cell biological program governed by *trh* will certainly merit future publication on its own.

II) The reviewers felt that a "toning down" of some of the language used in the text was called for, to better reflect just what the data was able to show. Much of this centered on properly interpreting the role of Trh, where two specific issues were raised:

• The *trh* mutant phenotype implies that Trh function is necessary for maintaining an invaginated structure, but stating that Trh mediates "tube formation" and "tube architecture" is not warranted. This (excessive) terminology appears throughout the text and in key section headings as well as the title of the paper. Alternative terms such as "morphogenesis" (Thus- "Two-step regulation of *trachealess* ensures tight coupling of cell fate with morphogenesis in *Drosophila* trachea" for the title) and "invaginated structure" rather than "tube"-containing terms should be substituted for the ones now used.

• Furthermore, *trh* expression may be turned on/maintained in the invaginated cells not because these cells somehow sense their new morphology, but because they have adopted a non-epidermal fate. This interpretation should be brought up and included when discussing the significance of the results.

---

## [Author Response]

Essential revisions:This is a valuable and interesting study whose primary message is the coupling of morphogenesis (in this case- invagination of surface epidermis) with the expression pattern of a key transcription factor that governs further tissue differentiation. All three reviewers viewed both the subject matter and the quality of much of the experimental work with high regard. All three, however, raised a number of non-trivial concerns regarding the significance of different results obtained, as well as aspects of the text, which seemed at times excessive in its interpretation of the data. The major critiques and suggested ways of addressing them include the following:I) The reported results can be divided conceptually into three groups-1) The role of Trh: the claim that Trh function is required only subsequent to invagination is a key observation at the heart of this study, which also contrasts with current "dogma" in the field. It is imperative therefore that a strong case be made.• Can the authors demonstrate that invagination followed by retraction is a prevalent phenotype in trh mutant embryos (i.e.- how often is it observed)?

We observed eight embryos at stage 12 and ten embryos at stages 15~16 in the heteroallelic combination of two TALEN alleles and the appearance of invaginated structures at stage 12 and their disappearance at stages 15~16, as shown in Figure 1F and G, respectively, with 100% penetrance. These TALEN alleles are supposed to be null, as described below. We have added this statement in the Results section (subsection “*trh* is required to maintain invaginated structures”, last paragraph).

• Can this be shown for full nulls (rather than/in addition to the hypomorphic trh^1^ allele)?

As we previously reported (Kondo et al., 2014), each TALEN allele has a frameshift mutation in the most upstream common exon among all isoforms. We predicted the amino acid sequences of the *trh^A14-3^* and *trh^B16-11^* mutants and demonstrated that both frameshift mutations lead to premature translational termination and the loss of the PAS A, PAS B and transcription activation domains in Figure 1—figure supplement 2A. The A14-3 allele also loses the bHLH domain. In addition, we showed that no anti-Trh signal was observed in the *trh^A14-3/B16-11^* mutants in Figure 1—figure supplement 2B. These results indicate that these alleles are null. We have shown that the heteroallelic combination of these two alleles replicated the phenotype observed with *trh^1^* (Figure 1D-G).

• Finally- the authors should explicitly address and discuss the discrepancies between their observations and those of previous studies in the Results or Discussion sections.

*The trh* mutant phenotype that we observed contradicted previous reports claiming that *trh* is required for invagination(Isaac and Andrew, 1996; Wilk et al., 1996). This conclusion was based mainly on the observation of a complete lack of tracheal structures and late tracheal markers in stage 15 or later. By looking closely into published data in the literature, we noted that signs of tracheal invagination are detectable in the early stage but are lost later in *trh* mutants (see, for example, Chung et al., 2011, Figure 2B, J, M, Figure 4B, E). The live imaging data in our present study provided definitive proof for the role of *trh* in the maintenance of the invaginated structures. We stated this point in the Discussion.

2) Upstream of trh: There was general agreement that the strong part of the study was the focus and analysis of how trh expression is regulated in invaginated vs. surface cells, and the identification of the R15F01 regulatory element as the key mediator of this program. Two additional experiments that could support and provide added significance to this notion were requested:• Assessment of whether driving trh expression using R15F01 alone is sufficient for rescue of the trh mutant phenotype.

*R15F01* activity is later and/or limited to a smaller number of trachealcells than *R14E10* (Figure 3—figure supplement 2). *trh*-overexpression by *R15F01-GAL4* in the *trh* mutants partially rescued tracheal morphogenesis (Figure 3—figure supplement 3). While the invagination anomaly of the *trh* mutants was not fully rescued at stage 12, *R15F01*-driven *trh* was able to maintain invaginated structures, although they showed an incomplete branching pattern, possibly due to a limited number of *R15F01-*positivecells. These results are consistent with the idea that *R15F01* activity is maintained in invaginated cells.

• Express GFP throughout the embryonic epidermis (e.g. via da-GAL4) via a UAS-based construct which also harbors R15F01, and examine whether GFP expression is silenced in the surface epidermal cells and maintained in invaginated cells of nascent salivary glands (an analogous tubular organ system). If this were to be the case, it would considerably strengthen the notion that the enhancing and silencing functions of R15F01 may indeed be sensitive to "tissue architecture", as claimed in the text.

We agree with the reviewer that this experiment is an important challenge, and we have performed additional experiments following the reviewer’s suggestion. We constructed 3xUAS and 3xUAS-*R15F01*-fused GFP reporters and integrated them into the same landing site (attP2). When we crossed them with arm-GAL4, which is an epithelial ubiquitous GAL4 driver, the 3xUAS-GFP reporter activity was detected ubiquitously with intense activity in surface epidermal and tubular hindgut cells. Since the reporter activity in salivary gland cells was relatively weak in our experimental setting, we focused on these two tissues. When we crossed the 3xUAS-*R15F01*-fused GFP reporter with arm-GAL4, epidermal GFP expression was repressed, while hindgut GFP expression was still detectable. Embryos harboring only the 3xUAS-*R15F01*-fused GFP reporter without arm-GAL4 did not show hindgut GFP expression, indicating that the hindgut activity was driven by GAL4. These results are also consistent with the notion that the silencing activity of *R15F01* is prominent in surface epidermal cells but is not functional in the cells of internal tubular organs relocated from the surface epidermis. We have added these results as Figure 5G-I.

3) Downstream of trh: Two of the reviewers found multiple problems and weaknesses in the sections on downregulation of Rho activity (possibly via Cv-c) and subsequent effects on actomyosin functions as a mechanism via which Trh governs tracheal morphogenesis. These related to the quality and interpretation of the data, and to the lack of coherent cellular and molecular scenarios that could account for formation of a tubular network. Given the extent of these criticisms, it appears premature to include this aspect of the study- certainly in its present state- in the manuscript, and the consensus among the reviewers is to remove it altogether. This would include the second and third sections of the Results. It is recommended, however, that demonstrations showing that invaginated tracheal cells have different expression or activation of markers than the surface epidermis be retained in some fashion, in order to support the notion of distinct fates. The authors could also speculate on possible downstream mechanisms on the basis of the Trh target list they compiled (Figure 3—figure supplement 2).This shortening of the text should not detract from the significance of the remaining portions of the study. Characterization of the cell biological program governed by trh will certainly merit future publication on its own.

We appreciate the constructive advice from the reviewers and agree that removing this part makes the paper simpler and strengthens the message of the paper. According to the reviewer’s suggestion, we decided to remove the section related to factors downstream of Trh and describe possible downstream mechanisms from the Discussion section. We will publish these results elsewhere in the future.

II) The reviewers felt that a "toning down" of some of the language used in the text was called for, to better reflect just what the data was able to show. Much of this centered on properly interpreting the role of Trh, where two specific issues were raised:• The trh mutant phenotype implies that Trh function is necessary for maintaining an invaginated structure, but stating that Trh mediates "tube formation" and "tube architecture" is not warranted. This (excessive) terminology appears throughout the text and in key section headings as well as the title of the paper. Alternative terms such as "morphogenesis" (Thus- "Two-step regulation of trachealess ensures tight coupling of cell fate with morphogenesis in Drosophila trachea" for the title) and "invaginated structure" rather than "tube"-containing terms should be substituted for the ones now used.

According to the reviewer’s suggestion, we have changed terms containing “tube” to “invaginated structures” or “morphogenesis”.

• Furthermore, trh expression may be turned on/maintained in the invaginated cells not because these cells somehow sense their new morphology, but because they have adopted a non-epidermal fate. This interpretation should be brought up and included when discussing the significance of the results.

We agree that escape from surface epidermis to the inside of the embryo could allow cells to acquire a non-epidermal fate instead of the forced induction of tracheal fate only in invaginated cells. Therefore, we further discussed the possible mechanisms to connect *trh* expression and epithelial structures in the Discussion section (subsection “Control mechanisms of *R15F01* CRM activity and Trh maintenance”, first paragraph).